# Vancomycin heteroresistance (hVISA) in MRSA links to treatment failure and supports a revised PAP-AUC threshold

Nikos Fatsis-Kavalopoulos [ORCID][1,4], Yong Kyun Kim[2,4], Yong Pil Chong [ORCID][3], Seongman Bae[3], So Yun Lim[3], Yang Soo Kim[3] ✉ & Dan I. Andersson [ORCID][1] ✉

Heteroresistance to vancomycin among methicillin-resistant *Staphylococcus aureus* (MRSA) remains a diagnostic and therapeutic problem in clinical microbiology. In this prospective cohort study of 842 adult patients with MRSA bacteremia in S. Korea, we investigate the prevalence, risk factors, and clinical implications of the heteroresistant vancomycin-intermediate *S. aureus* (hVISA) phenotype. The hVISA phenotype is detected in 22% of cases. Multi-variable regression analysis reveals strong positive associations between hVISA and hospital-acquired infection, prior anti-MRSA therapy, vancomycin exposure, and particularly vancomycin MIC (odds ratio 15.2 per 1 mg/L increase, p < 0.001). Strikingly, patients infected with hVISA strains have a lower 90-day mortality compared to those with fully susceptible strains (hazard ratio 0.66, p = 0.019), suggesting a possible trade-off between resistance and virulence. However, in hVISA strains treated with vancomycin, outcomes reverse: mortality more than doubled (HR 2.5, p < 0.001), bacteremia persisted longer, and relapse rates increased fivefold. Using maximally selected rank statistics, we identify a PAP–AUC threshold of 0.65 as the first clinically derived breakpoint predictive of mortality risk, providing an actionable definition of vancomycin heteroresistance. These findings underscore the clinical relevance of hVISA, and support routine testing for heteroresistance to inform treatment decisions.

Methicillin-resistant *S. aureus* (MRSA) remains one of the most formidable challenges in clinical microbiology and infectious disease management. Identified as a priority pathogen by the World Health Organization (WHO)[1], MRSA is a leading cause of healthcare-associated infections worldwide[2,3], associated with severe morbidity[3–5], prolonged hospital stays[5,6], and increased mortality[3–5,7]. The continued emergence of MRSA strains with reduced susceptibility to frontline antibiotics poses a significant threat to public health, prompting the need for continuous surveillance and the development of effective therapeutic strategies[8].

Among the diverse resistance mechanisms in MRSA, one that has garnered increasing attention is heteroresistance, a phenotype where a bacterial population contains a small subpopulation of cells with significantly higher resistance levels compared to the susceptible majority[9,10]. In the context of MRSA, this phenomenon has been most notably observed as heteroresistant vancomycin (VAN)-intermediate *S. aureus* (hVISA)[11,12]. The hVISA phenotype is particularly concerning because it erodes the efficacy of VAN, which remains the cornerstone of treatment for severe MRSA infections, especially bloodstream

[1]Department of Medical Biochemistry and Microbiology, Uppsala University, Uppsala, Sweden. [2]Division of Infectious Diseases, Department of Internal Medicine, Hallym University Sacred Heart Hospital, Hallym University College of Medicine, Seoul, South Korea. [3]Division of Infectious Diseases, Department of Medicine, Asan Medical Center, University of Ulsan College of Medicine, Seoul, South Korea. [4]These authors contributed equally: Nikos Fatsis-Kavalopoulos, Yong Kyun Kim. ✉e-mail: yskim@amc.seoul.kr; Dan.Andersson@imbim.uu.se

infections[13] and infective endocarditis[14,15]. As VAN is often the last line of defence against MRSA, the emergence of hVISA threatens to diminish one of the most valuable therapeutic options available.

The hVISA phenotype was first described in Japan in 1997[12], when isolates from patients with persistent MRSA bacteremia demonstrated intermediate resistance to VAN despite being classified as susceptible by standard MIC testing. Since then, the clinical implications of hVISA have remained a topic of debate[16]. The phenotype is defined microbiologically by the population analysis profile–area under the curve (PAP–AUC) method, where the test isolate's AUC relative to the reference strain (Mu3, ATCC 700698) is ≥0.9[11]. This method is considered the gold standard, but it is labour-intensive and time-consuming, limiting its routine use. In addition, and importantly, the AUC cutoff value for hVISA has no established link to treatment outcome, limiting its usefulness in clinical settings.

Several studies have identified key risk factors associated with the development of the hVISA phenotype, primarily linked to antibiotic exposure and patient comorbidities. The most consistently reported factor is prior VAN use, which significantly increases the likelihood of hVISA emergence[17,18]. Other studies have also highlighted chronic renal failure, prolonged hospitalization, and immunosuppression as important predictors, likely due to increased antibiotic pressure and reduced host defenses[19,20]. Additionally, hospital-acquired infections and higher VAN MICs have been linked to hVISA, reflecting the role of healthcare settings and intense antibiotic use in fostering heteroresistance[21,22].

Despite over two decades of research, there remains no consensus on the clinical relevance of hVISA. Some studies have linked the phenotype to increased morbidity, prolonged bacteremia, and treatment failure[17,19,23–27], while others have found no significant impact on patient outcomes, including mortality[18,22,23,28,29]. The inconsistency between clinical trials and observational studies has made it difficult to draw definitive conclusions about whether hVISA should be regarded as a marker of poor prognosis.

One possible reason for the variability in reported outcomes is the way hVISA is studied. Some investigations have focused solely on the phenotype itself, assessing its presence and correlating it with clinical outcomes, while others have examined the interaction between hVISA and VAN therapy, analyzing whether the combination of heteroresistance and VAN treatment influences outcomes differently. This duality in study designs has contributed to divergent findings and conflicting interpretations, making it challenging for clinicians to determine the true clinical impact of hVISA.

In this study, we aimed to address these gaps by conducting a large prospective cohort analysis to assess both the prevalence and clinical implications of the hVISA phenotype in MRSA bacteremia. Uniquely, we adopted a dual approach to examine not only the independent effects of the phenotype but also its interaction with VAN therapy. Furthermore, we identified potential risk factors associated with hVISA emergence and determined an AUC breakpoint for increased risk of mortality, providing clinicians with evidence to guide early identification and more tailored therapeutic strategies.

## Results

### Prevalence and risk factors of hVISA

During the study period, we initially identified 1592 patients with a first episode of MRSA bacteremia. After applying predefined exclusion criteria, including polymicrobial bacteremia ($n = 207$) and discharge prior to confirmation of a positive blood culture ($n = 57$), a total of 1328 patients remained for the final cohort. Among these, 842 patients with MRSA bloodstream isolates who underwent PAP testing to determine hVISA status met the inclusion criteria. Of these, 184 isolates were identified as hVISA, yielding a prevalence of 22% in the study cohort (characteristics of which are shown in Table 1).

Multivariable logistic regression was used to assess associations between clinical variables and the hVISA phenotype, aiming to identify potential patient-level risk factors (Fig. 1). An interaction term between age and sex was included to evaluate whether the association between age and hVISA differed by sex, and all associated Odds ratios and $p$ values are shown in Table 2.

The multivariate analysis indicated that patients with community-acquired MRSA bacteremia had approximately half the odds of exhibiting the hVISA phenotype compared with those with nosocomial infections (OR 0.5, 95% CI 0.3–0.8; $p = 0.008$), indicating that hospital acquired infections are a significant risk factor for hVISA. Similarly, prior anti-MRSA therapy was associated with a twofold increase in the odds of hVISA (OR 2.0, 95% CI 1.2–3.4; $p = 0.006$), and previous VAN exposure was also linked to higher odds of the phenotype (OR 1.7, 95% CI 1.1–2.6; $p = 0.024$). VAN MIC by broth microdilution (BMD) was the strongest independent predictor of hVISA, with each mg/L increase in MIC associated with a more than 15-fold increase in the odds of the phenotype (OR 15.3, 95% CI 7.8–30.9; $p < 0.001$). Further analysis of the distribution of VAN MIC values (Supplementary Fig. 1) indicates a significantly different distribution between the hVISA and VSSA strains (t test $p = 0.001$), reiterating the association between VAN MIC and the hVISA phenotype.

Out of the 45 $S.$ $aureus$ sequence types present in this cohort, 7 had significant changes in relative risk of producing hVISA phenotypes (Table 3 and supplementary information MLST relative risks). The greatest increase in relative risk was found in ST5 with a 13 fold increase in the risk of a strain of that Sequence Type being hVISA (compared to the rest of the cohort, $\chi^2$ test, 95% CI: 0.3500–0.9031, $p = 0.003$). In contrast, ST239 had the greatest fold reduction in risk with a 25-fold reduction in relative risk of a strain of that Sequence type being hVISA (compared to the rest of the cohort, $\chi^2$ test, 95% CI: 0.007959–0.2536, $p = 0.001$).

### Effects of the hVISA phenotype on clinical outcomes

Multivariable analyses, as detailed in the Methods section, were used to assess associations between the hVISA phenotype and clinical outcomes. As seen in Table 4, patients with hVISA bacteremia had significantly lower 90-day all-cause mortality compared with those

**Table 1 | Characteristics of hVISA and non hVISA subcohorts, age, CCI (=Charlson Comorbidity Index) score, McCabe score, duration of SAB, Pitt score, sepsis grade are noted as median and range**

| | hVISA patients (184) | non hVISA patients (658) |
|---|---|---|
| Age | 63.0 (20.0 92.0) | 65.0 (19.0–98.0) |
| Female | 24% (44) | 39% (256) |
| Male | 76% (140) | 61% (402) |
| CCI score | 3.0 (0.0–9.0) | 3.0 (0.0–13.0) |
| McCabe score | 3.0 (1.0–3.0) | 3.0 (1.0–3.0) |
| Survival within 90 days from positive index culture | 68% (125) | 70% (460) |
| Duration of SAB | 1.0 (1.0–68.0) | 1.0 (1.0–74.0) |
| Recurrence of SAB within 90 days from positive index culture | 6% (12) | 5% (32) |
| Pitt score | 1 (0.0–18.0) | 1.0 (0.0–13.0) |
| Sepsis grade | 2.0 (1.0–4.0) | 2 (1.0–4.0) |
| Admission in the ICU | 31% (57) | 19% (125) |
| VAN treatment | 80% (147) | 82% (539) |
| Previous VAN treatment | 34% (62) | 16% (105) |
| Previous any AB treatment | 82% (150) | 57% (375) |

Gender, mortality within 90 days from positive index culture, recurrence of SAB within 90 days from positive index culture, admission to the ICU, and antibiotic treatments are noted as percentage of patients and number.

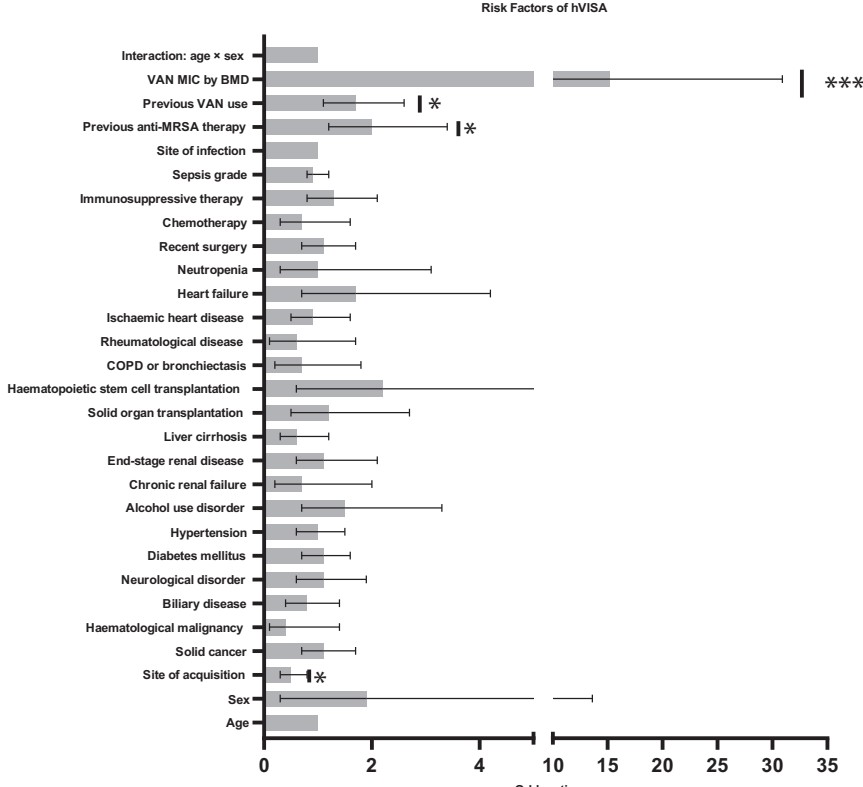

**Fig. 1 | Odds ratios of Risk factors associated with hVISA.** Shown as value with 95% CI (error bars denote CI), * denotes 0.001<*p* < 0.05 and *** denotes *p* < 0.001 (multivariate logistic regression). *p* = 0.0001 for VAN MIC by BMD, *p* = 0.024 for Previous VAN use, *p* = 0.006 for Previous anti MRSA therapy, and *p* = 0.081 for Site of acquisition.

infected with non-hVISA MRSA strains. Thus, in multivariable Cox regression analysis, the hVISA phenotype was surprisingly associated with a 34% reduction in the hazard of death within 90 days of admission, (HR 0.66, 95% CI 0.47–0.94; *p* = 0.019).

### Effects of hVISA phenotype in patients with VAN therapy

A combination metric was included to assess whether the effect of hVISA on the clinical outcomes was modified by receipt of VAN as definitive therapy, i.e., the effect of hVISA on clinical outcomes when treated with VAN. This analysis aimed to determine whether VAN use altered the association between the hVISA phenotype and clinical endpoints (Table 5).

The presence of the hVISA phenotype in combination with VAN therapy was associated with significantly worse clinical outcomes across several endpoints as compared to all other patients in the cohort. Most notably, the hazard of all-cause mortality within 90 days of positive index culture in the hVISA group treated with VAN is more than 2 fold higher than for the rest of the patients (HR 2.5, 95% CI 1.76–3.53; *p* < 0.001). Additionally, patients in this group had a significantly prolonged duration of bacteremia, with an estimated increase of 1.8 days compared to other groups (95% CI 0.2–3.5; *p* = 0.024). Recurrence of MRSA bacteremia within 90 days from positive index culture was also more likely, with 23% higher odds observed in the hVISA–VAN group (OR 1.23, 95% CI 1.03–1.86; *p* = 0.050).

### AUC hVISA BreakPoint is associated with significantly increased risk of mortality

Using maximally selected rank statistics, we identified a PAP–AUC threshold of 0.65 as the optimal cut-off for stratifying 90-day all-cause mortality risk in patients with MRSA bacteremia. Patients with isolates exceeding this threshold had a significantly higher hazard of death

within 90 days compared to those with lower AUC values. The result was statistically significant (*p* = 0.043), indicating that this threshold has prognostic relevance.

### Discussion

In this study, we aimed to investigate the prevalence and clinical implications of the hVISA phenotype within a large patient cohort. To achieve this, we conducted a dual analysis that examined both the independent effects of the hVISA phenotype and its interaction with VAN treatment. This approach was designed to disentangle potential associations between the phenotype itself and adverse clinical outcomes from those arising due to inappropriate treatment of HR infections[30].

One of the major strengths of our study is its scale, making it the largest study to date examining the clinical impact of the hVISA phenotype in MRSA bacteremia. We enrolled 842 patients over several years, significantly surpassing both the size of previous studies. This large cohort allowed us to investigate a broad range of more than 30 predictors and risk factors for the hVISA phenotype, and to analyze associations with several clinical outcomes, thereby enhancing the generalizability of our findings. Additionally, all MRSA isolates were retrospectively PAP tested, ensuring consistent and accurate identification of the hVISA phenotype.

Our study had additional key features setting it apart from previous investigations, beyond its extensive cohort size. One of the most important advantages is that not all patients in the cohort were treated with VAN, specifically 18% (153 patients) did not receive VAN. This allowed for a unique dual analysis, enabling us to examine both the potential of the hVISA phenotype itself to influence clinical outcomes and the increased risk of worse outcomes when the phenotype was treated with VAN. This framework allowed us to disentangle the direct effects of the hVISA phenotype from the specific impact of its

**Table 2 | Associations of clinical variables to a patient having an hVISA infection**

| Clinical variables/ comorbidities | P value | Odds ratio (95% CI) |
|---|---|---|
| Age | 0.541 | 1.0 (1.0–1.0) |
| Sex | 0.532 | 1.9 (0.3–13.6) |
| Site of acquisition | 0.008 | 0.5 (0.3–0.8) |
| Solid cancer | 0.684 | 1.1 (0.7–1.7) |
| Hematological malignancy | 0.176 | 0.4 (0.1–1.4) |
| Biliary disease | 0.369 | 0.8 (0.4–1.4) |
| Neurological disorder | 0.747 | 1.1 (0.6–1.9) |
| Diabetes mellitus | 0.805 | 1.1 (0.7–1.6) |
| Hypertension | 0.902 | 1.0 (0.6–1.5) |
| Alcohol use disorder | 0.263 | 1.5 (0.7–3.3) |
| Chronic renal failure | 0.540 | 0.7 (0.2–2.0) |
| End-stage renal disease | 0.778 | 1.1 (0.6–2.1) |
| Liver cirrhosis | 0.173 | 0.6 (0.3–1.2) |
| Solid organ transplantation | 0.657 | 1.2 (0.5–2.7) |
| Haematopoietic stem cell transplantation | 0.260 | 2.2 (0.6–8.7) |
| COPD or bronchiectasis | 0.440 | 0.7 (0.2–1.8) |
| Rheumatological disease | 0.338 | 0.6 (0.1–1.7) |
| Ischemic heart disease | 0.692 | 0.9 (0.5–1.6) |
| Heart failure | 0.245 | 1.7 (0.7–4.2) |
| Neutropenia | 0.993 | 1.0 (0.3–3.1) |
| Recent surgery | 0.654 | 1.1 (0.7–1.7) |
| Chemotherapy | 0.384 | 0.7 (0.3–1.6) |
| Immunosuppressive therapy | 0.259 | 1.3 (0.8–2.1) |
| Sepsis grade | 0.640 | 0.9 (0.8–1.2) |
| Site of infection | 0.254 | 1.0 (1.0–1.0) |
| Previous anti-MRSA therapy | 0.006 | 2.0 (1.2–3.4) |
| Previous VAN use | 0.024 | 1.7 (1.1–2.6) |
| VAN MIC by BMD | <0.001 | 15.2 (7.8–30.9) |
| Interaction: age × sex | 0.865 | 1.0 (1.0–1.0) |

*COPD* chronic obstructive pulmonary disease (multivariate logistic regression).

**Table 3 | Sequence types (STs) associated with significant increases in Relative Risk of being an hVISA compared to the rest of the cohort (two-sided $\chi^2$ test)**

| ST | Non-hVISA | hVISA | Relative Risk | CI | p value |
|---|---|---|---|---|---|
| ST239 | 89 | 1 | 0.04537 | 0.007959–0.2536 | 0.0001 |
| ST72 | 4 | 14 | 11.7 | 4.093–33.47 | 0.0001 |
| ST8 | 95 | 7 | 0.2905 | 0.1385–0.5985 | 0.0004 |
| ST30 | 50 | 2 | 6.568 | 1.799–24.41 | 0.0019 |
| ST34 | 199 | 89 | 1.404 | 1.135–1.724 | 0.002 |
| ST5 | 1 | 3 | 13.63 | 1.961–94.67 | 0.003 |
| ST188 | 51 | 21 | 0.5583 | 0.3500–0.9031 | 0.0173 |

treatment with VAN, addressing a major limitation in previous studies that often conflated these effects.

By implementing this dual analysis, we determined that poor clinical outcomes were not inherently driven by the hVISA phenotype itself, but rather by the use of vancomycin as treatment for vancomycin-heteroresistant strains. This distinction is particularly important as it challenges the notion that the phenotype alone predicts adverse outcomes, instead highlighting that the therapeutic approach itself significantly influences prognosis.

Furthermore, our analysis of the impact of VAN treatment on hVISA strains was conducted using multivariate models that accounted

**Table 4 | Associations of hVISA to clinical outcomes, HR is hazard ratio, OR is odds ratio and β is regression coefficient (multivariate Cox hazard regression)**

| Outcome | Effect size (95% CI) | P value |
|---|---|---|
| Mortality within 90 days from positive index culture | 0.66 (HR) [0.47–0.94] | 0.019 |
| Duration of bacteremia | −0.83 (β) [−2.24–0.58] | 0.248 |
| Recurrence of SAB within 90 days from positive index culture | 0.99 (OR) [0.99–1] | 0.99 |
| Sepsis grade | 0.04 (β) [−0.12–0.20] | 0.635 |
| ICU admission | 1.02 (OR) [0.63–1.64] | 0.935 |

**Table 5 | Associations of hVISA treated with VAN to clinical outcomes, HR is hazard ratio, OR is odds ratio and β is regression coefficient (multivariate Cox hazard regression)**

| Outcome | Effect size (95% CI) | P value |
|---|---|---|
| Mortality within 90 days from positive index culture | 2.5 (HR) [1.76–3.53] | 0.001 |
| Duration of bacteremia | −1.8 (β) [−3.47––0.19] | 0.024 |
| 90-day recurrence of SAB among patients surviving 90 days from positive index culture. | 5.05 (OR) [1.63–22.64] | 0.013 |
| Sepsis grade | 0.12 (β) [−0.06–0.31] | 0.187 |
| ICU admission | 0.93 (OR) [0.49–1.81] | 0.824 |

for increased MIC of VAN as a potential confounding factor. This approach ensured that the associations we identified between VAN treatment and poor outcomes are independent of the effects of elevated MICs, thereby isolating the true impact of treating heteroresistant strains with VAN. This allowed us to accurately attribute associations to adverse outcomes, specifically to the treatment of hVISA with VAN, rather than conflating them with the consequences of treatment failure due to higher MICs of VAN alone. This distinction is critical as it provides clearer insights into the clinical challenges of managing hVISA.

In this cohort of 842 patients with MRSA bacteremia, 184 patient isolates of MRSA (22 %) exhibited the hVISA phenotype. Among 30 demographic, clinical, and microbiological potential risk factors, hVISA remained most strongly linked to four factors. Patients who acquired their infection in the hospital were twice as likely to harbour hVISA as those infected in the community. Patients who had already received an anti-MRSA antibiotic had an approximately 2-fold increased risk of hVISA. Similarly, previous exposure to VAN increased the likelihood of hVISA by roughly 70 %. The single strongest predictor was the MIC of VAN: for each 1 mg/L stepwise increase in MIC, the probability of encountering hVISA increased more than fifteen-fold. These findings are corroborated by smaller-scale studies[17,18] that have detected associations between the emergence of hVISA and prior exposure to VAN. This implies that antibiotic exposure history, particularly with VAN, is a central driver of hVISA development, even though the role of patient comorbidities may vary between different healthcare settings and populations.

When examining the risks associated with the hVISA phenotype alone, we found one clear and unexpected signal: patients with hVISA were about one-third less likely to die within 90 days of positive index culture than those infected with VAN-susceptible MRSA. One possible explanation for the reduced mortality could be that hVISA strains are less virulent than the VAN-susceptible ones[22,31]. Heteroresistance often arises through thickened cell-wall architecture[32–34] and altered autolysis[32,35], adaptations that can help a minority subpopulation survive antibiotic exposure but impose a biological cost[36]. Mechanistic studies on S. *aureus* heteroresistance[37] have shown that that cost often

takes the form of reduced replication rates. Consequently, the slower growth and replication of hVISA may allow the host immune system to more efficiently clear the infection, leading to increased survival. Notably, the increased survival in patients with hVISA infections has been previously observed in a study of 401 MRSA isolates, which specifically highlighted that hVISA, when compared to VSSA in a large cohort of patients, was paradoxically associated with lower mortality as well[22].

Several MLSTs in our cohort were shown to be associated with significantly changed Relative Risk of exhibiting the hVISA phenotype. Although in the context of this study, we have documented them as associations, we believe these observations carry diagnostic potential as an early warning sign that could trigger a full phenotypic examination of a strain for hVISA behaviour. Further mechanistic investigations in the relationship between those MLSTs and the hVISA phenotype could also potentially lead to important insights on the causes of VAN heteroresistance in MRSA.

In this cohort, both spiral plating and PAP testing demonstrated reduced reproducibility and concordance with each other, with disagreements between the two methods or replicates exceeding 30%. This clearly illustrates that clinical testing for hVISA is in great need of standardisation and perhaps future dedicated heteroresistance diagnostics[38].

Smaller scale studies[18–20,28,29] have identified associations between hVISA and higher rates of treatment failure as well as prolonged bacteremia, but they did not consistently find an increase in mortality, aligning with our observation. An increase in mortality was reported specifically when hVISA coexisted with β-lactam-induced VAN resistance (BIVR)[24], a factor not evaluated in our cohort. It is of note that potential differences in MRSA strain types and our study's inclusion of both community and hospital-acquired infections offer a possible explanation for further discrepancies between the results.

In contrast, in this study, when the hVISA phenotype coincided with receipt of VAN as therapy, the clinical picture reversed, consistent with a previous smaller study[21]. Patients with hVISA who remained on VAN had more than twice the hazard of death 90 days after positive index culture and an average prolongation of bacteremia by almost two days. Notably, a two-day extension brings the duration of bloodstream infection to the cusp of definitions of "persistent" bacteremia (often defined as $\geq 2$–7 days)[17,39]. Patients who survived 90 days from positive index culture had roughly 5 times higher odds of relapse within the same time span.

Our findings regarding the combined effect of the hVISA phenotype and VAN treatment are consistent with several smaller studies that reported significantly worse clinical outcomes or even treatment failure[21] when heteroresistant S. aureus strains were treated with VAN. hVISA bloodstream infections treated with VAN were observed[17] to have higher treatment failure rates compared to VSSA, driven by persistent bacteremia and relapse. Similarly, in patients with MRSA infective endocarditis[25] and pneumonia[27], hVISA treated with VAN was associated with significantly higher failure and mortality rates. Patients in an ICU setting with hVISA[26] had significantly higher in-hospital mortality when treated with VAN. There are, however, differences between studies that arise from variations in clinical settings (ICU vs. non-ICU), VAN dosing strategies, and differences in how VAN MICs are measured, which can significantly influence treatment outcomes.

It is important to consider the limitations that temper the generalizability of this study's conclusions. First, this investigation is done as a single-centre study undertaken in a large tertiary hospital in Seoul, so local prescribing practices, pathogen ecology and infection-control measures may not reflect those of other regions or levels of care. Second, the timespan of data collection (2008–2023) is long, and during this time VAN dosing strategies, breakpoint definitions and the availability of alternative anti-MRSA agents have evolved. Thus, such changes in clinical practice over time may have biased both prevalence estimates and outcome associations. Mitigating this limitation is the fact that the cohort was assembled at a large tertiary-care hospital that serves as a national referral center, receiving patients from across the country. This broad catchment area supports the view that the study population is likely to be nationally representative. Furthermore, the data were derived from a prospective observational cohort built around formal Infectious Diseases consultations for SAB, ensuring systematic case identification and clinical assessment; however, TDM was not applied in all patients, so inconsistencies in VAN dosing cannot be ruled out as a confounder of worse patient outcomes. Third, the observational design precludes firm causal inference and is susceptible to residual confounding, particularly confounding by indication, as the choice to prescribe or not prescribe VAN to a patient is subject to the prescribing clinician's judgment. It is also important to note that although the PAP–AUC reference method was used to classify the isolates, hVISA detection relied on stored isolates, raising the possibility of phenotype instability or laboratory misclassification. Finally, we captured only the first episode of bacteremia, potentially overlooking patients with multiple infections.

Additional insights into the nature of the hVISA phenotype, as well as the mechanisms that lead to it, could be gleaned from future clinical studies of a more narrow and focused scope. Assembling a clinical cohort where all MRSA isolates are whole-genome sequenced could enable genome-wide association analyses that were not possible in this current study. Additionally, studies that assemble cohorts of hVISA patients treated with other antibiotics can, together with in vitro experiments on the growth rate of the patient isolates, shed more light into the mechanisms behind the association of hVISA with lower mortality rates when not treated with VAN.

The findings of this study carry several important implications for clinical practice concerning hVISA. Considering our findings in the context of the previous observation of VAN treatment for hVISA strains, it would imply that VAN exposure fails to adequately eradicate heteroresistant subpopulations. Furthermore, we identified that hospital-acquired infections and prior antibiotic treatment are significant risk factors for the emergence of hVISA, consistent with observations from smaller studies. This suggests that clinical practices themselves may inadvertently contribute to the development of S. aureus VAN heteroresistance, indicating that changes in hospital antibiotic stewardship and infection control protocols might be warranted. As expected, the most significant risk factor for hVISA is the MIC of VAN itself, suggesting that heteroresistance may serve as a precursor to fully developed resistance[40,41].

One of the most clinically significant results of our study was the identification of a new Break Point for the PAP–AUC values used for the identification of hVISA. Using maximally selected rank statistics, we determined that a PAP–AUC value of 0.65 was the optimal cut-off for stratifying risk of 90-day mortality ($p = 0.045$). Patients with isolates exceeding this threshold had a significantly higher risk of death, independent of the MIC of VAN. This finding introduces, for the first time, a clinically derived and outcome-based breakpoint for PAP–AUC, grounded in a large real-world cohort. It should be noted that the original PAP-AUC cut-off value of $\geq 0.9$[11] was based on a much smaller number of isolates, and importantly, without any linkage to clinical outcome. If validated in external datasets, this new threshold could shift how hVISA is detected, interpreted, and managed in clinical microbiology and infectious disease practice.

In conclusion, our results indicate that undiagnosed VAN heteroresistance poses a direct threat to patient safety and represents an additional burden on healthcare systems. Given these findings, we suggest that integration of specialised diagnostic tests into routine clinical practice to enable rapid detection of this heteroresistance can help prevent treatment failures and mitigate complications linked to heteroresistance.

## Methods

### Study population

We conducted a prospective observational cohort study at Asan Medical Center, a 2700-bed tertiary-care referral hospital in Seoul, South Korea. Our cohort of MRSA bacteremia was derived from the largest tertiary-care hospital in South Korea, which serves as a national referral centre and therefore provides a population broadly representative of cases across the country. In addition, we analyzed data from a prospective observational cohort that was established around formal Infectious Diseases consultations for MRSA bacteremia, thereby ensuring systematic case identification and consistent clinical assessment. We included patients aged 18 years or older with at least one blood culture positive for *S. aureus* between Aug 1, 2008, and Apr 30, 2023, whose corresponding isolates underwent PAP testing to determine hVISA status. Individuals with polymicrobial bacteremia, multiple episodes, or those discharged before confirmation of a positive blood culture were excluded to preserve independent observations. All MRSA isolates from the first day of bacteremia were stored frozen at −80 °C for further analysis. This study was approved by the Institutional Review Board of Asan Medical Center (IRB No. 2013-0234),with informed consent of all patients. All patient data is available in the source data file.

### Identification of hVISA phenotype

All stored MRSA isolates were tested for the hVISA phenotype using either the PAP–AUC method[11] or spiral plate testing, or both. Testing was performed in at least two replicates. In cases of classification disagreement between replicates, a third replicate was conducted, and the final classification was based on the two agreeing results. For patients where both methods were performed but gave discordant classifications, the spiral plate result was used. For AUC-based analyses, the value used was the average of the two concordant replicates —whether from PAP–AUC, spiral plate, or both if all agreed. All testing was performed retrospectively, and clinicians were blinded to hVISA status. AUC data is available in the source data file. Out of 65 VSSA isolates tested with both methods 30% (20) show disagreement between methods and 70% (45) agreement. Out of 31 hVISA tested with both methods 38% (12) show disagreement between the methods 61% (19) agreement.

### Microbiological data

Isolates were assessed for antimicrobial susceptibility using the MicroScan system (Dade Behring, CA, USA), following CLSI M100 (31st ed.). Methicillin resistance was confirmed by *mecA* PCR[42]. VAN minimum inhibitory concentrations (MICs) were determined via BMD and Etest in a research laboratory; these results were not available to clinicians. All microbiological data is available in the source data file.

### Outcomes

The primary outcome was all-cause mortality within 90 days of the index positive culture. Secondary outcomes included duration of *S. aureus* bacteremia (SAB) (defined as duration while on active antibiotic therapy[39,43], ICU admission, sepsis grade at presentation (defined according to consensus criteria[44], and SAB recurrence within 90 days of the index positive culture among patients who survived.

### Statistical analysis and study design

We used multivariable Cox proportional hazards regression to assess 90-day mortality. Logistic regression was applied to SAB recurrence, risk factors of hVISA analysis and ICU admission; linear regression was used for duration of bacteremia and sepsis severity. All models were adjusted for potential confounders, including demographic variables, site of infection acquisition, comorbidity conditions, severity indices and microbiological characteristics as seen in Table 6. An interaction term between hVISA and VAN treatment was included to evaluate the

**Table 6 | List of Risk Factors and multivariate model parameters**

| Risk Factors | Multivariate model confounders |
|---|---|
| Age | Age |
| Sex | Sex |
| Site of acquisition | Site of acquisition (healthcare-associated Vs nosocomial) |
| Solid cancer | |
| Hematological malignancy | Solid cancer |
| Biliary disease | Hematological malignancy |
| Neurological disorder | Biliary disease |
| Diabetes mellitus | Neurological disorder |
| Hypertension | Diabetes mellitus |
| Alcohol use disorder | Hypertension |
| Chronic renal failure | Alcohol use disorder |
| End-stage renal disease | Chronic renal failure |
| Liver cirrhosis | End-stage renal disease |
| Solid organ transplantation | Liver cirrhosis |
| Haematopoietic stem cell transplantation | Solid organ transplantation |
| COPD or bronchiectasis | Haematopoietic stem cell transplantation |
| Rheumatological disease | COPD or bronchiectasis |
| Ischemic heart disease | Rheumatologic disease |
| Heart failure | Ischemic heart disease |
| Neutropenia | Heart failure |
| Recent surgery | McCabe score |
| Chemotherapy | Charlson index |
| Immunosuppressive therapy | Neutropenia |
| Sepsis grade | Recent surgery |
| Site of infection | Chemotherapy |
| Previous anti-MRSA therapy | Mucositis |
| Previous VAN use | Immunosuppressive therapy |
| VAN MIC by BMD | ICU care |
| Interaction: age × sex | Previous anti-MRSA therapy |
| | Previous vancomycin exposure |
| | Vancomycin MIC by BMD |

*COPD* chronic obstructive pulmonary disease.

modification of outcomes by therapy. All raw output data of the statistical analysis is available in the source data file.

### Reporting summary

Further information on research design is available in the Nature Portfolio Reporting Summary linked to this article.

## Data availability

All primary data is available in the source data file. Source data are provided with this paper.

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

## Acknowledgements
This work was supported by grants from the Knut and Alice Wallenberg Foundation (2018.0168), and the Swedish Research Council to D.I.A. (2021-02091). The authors would like to acknowledge the Uppsala Antibiotic Center (UAC) for all its support. The funders of the study had no role in study design, data collection, data analysis, data interpretation, or writing of the report.

## Author contributions
Study conceptualization: N.F.-K. and Y.K.K. Formal Analysis: N.F.-K. and Y.K.K. Funding acquisition: D.I.A. Investigation: N.F.-K., Y.K.K., Y.P.C., S.B., S.Y.L., Y.S.K., and D.I.A. Writing of original draft: N.F.-K. and Y.K.K. Review, editing and data verification of final draft: N.F.-K., Y.K.K., Y.P.C., S.B., S.Y.L., Y.S.K., and D.I.A. Access to raw clinical data: Y.K.K. and Y.S.K.

## Funding

## Competing interests
The authors declare no competing interests.
