## [Transparent Peer Review file · Nature Communications]

Vancomycin Heteroresistance (hVISA) in MRSA links to treatment failure and supports a revised PAP-AUC threshold

Corresponding Author: Professor Dan Andersson

Version 0:

Reviewer comments:

Reviewer #1

(Remarks to the Author)

The authors of "Vancomycin heteroresistant (hVISA) MRSA infections: risk factors, effects on vancomycin therapy and a clinically derived threshold for its detection" report a study exploring the clinical relevance of the hVISA phenotype. They collected clinical and laboratory data from 842 patients with MRSA bacteremia and using single and multivariate analyses explored if hVISA was correlated with worse disease outcomes – with a focus on disentangling the impact of hVISA alone from the variety of other features that contribute to disease outcome. Further, the authors explored what features were most predictive of hVISA strains, which could possibly be used as a diagnostic marker given that confirmatory lab testing of hVISA using reference methods remains a labour-intensive.

The manuscript is well written and succinct – almost too succinct - and inclusion of more detail could improve the text (see below). The laboratory methods for detection of hVISA and the statistical analysis is appropriate. Project ethics have been obtained. The data and detailed analysis parameters have been included in Supp Table 1. The manuscript was provided without line numbers.

Main comments

More detail is needed in the methods/results to understand the study hospital and therefore the context of the reported findings. In table 2, the 90-day mortality is reported as 68% and 70% for hVISA/MRSA cohorts. Is this correct? This value is high compared to a number of studies of MRSA bacteraemia (meta-analysis doi:10.1016/j.cmi.2022.03.015). Can the authors please provide more detail about the patient population included in the study and a potential explanation for these high values. Further, hVISA has often been linked with persistent bacteraemia. So how is "duration of bacteraemia" defined in the study hospital? Please explain. I would also recommend (in table 2) providing the full ranges for the values alongside or in place of the 25/75 percentiles.

The authors have not provided any information or discussion on the role of bacterial genetics in influencing clinical outcomes (other than briefly that hVISA has been linked with reduced disease severity). Although linking key bacterial genotypes with clinical outcomes is an emerging area of research – there is a number of papers that have correlated the hVISA phenotype with key mutations (reviews doi: 10.1016/j.jgar.2014.04.006 & 10.1128/CMR.00042-09), and also demonstrated that certain MRSA lineages, such as the HA-MRSA lineage ST239, has evolved towards higher VAN MICs (doi: 10.1128/mBio.00080-15). While a lot remains unknown in this area, the literature strongly suggest that some MRSA lineages have a greater predisposition to become hVISA. Therefore, understanding local molecular epidemiology of MRSA is important for understanding risk factors for hVISA. The authors have captured information on MLST for all isolates (Supp Table) but do not appear to have considered this feature in their analyses. An analysis that looks into this feature, or at a minimum, explanation of why it has not been considered should be added.

In terms of VAN MICs, please include more information – or even a graph/table – to describe the distribution of MIC from the hVISA and MRSA cohorts. Even an ecological cut-off value (or similar) for the hVISA phenotype in this study would be useful.

Minor changes

1. At the start of the introduction, please add the MRSA acronym for consistency with other acronym use
2. In the methods, please provide the name of the funding body that supported this work.
3. Table 3 – I suggest showing the OR information in a graph format
4. In Supp Table 1 – (in AUC tab) the results for the PAP are recorded under “Modified PAP” – if modification have been made the reference PAP procedure, please described these changes in the methods. If not, please amend the column header.

Reviewer #2

(Remarks to the Author)

Thank you for the opportunity to review the manuscript, “Vancomycin heteroresistant (hVISA) MRSA infections: risk factors, effects on vancomycin therapy and a clinically derived threshold for its detection.” Vancomycin is now the first-line therapy for MRSA bacteremia, and it is essential that we can identify any subgroup of patients with this high-mortality infection who have poorer outcome with vancomycin as alternative therapies are available. The authors of the present a report on 842 adult patients with MRSA bacteremia in South Korea, and for each they collected extensive clinical data and assessed the presence of an hVISA isolate. They compared hVISA with non-hVISA infections to assess clinical differences in the patients presenting with infection and then its impact on clinical outcomes. Among the 842 MRSA isolates, the authors found that 22% were hVISA by population analysis profile (PAP) or by spiral gradient profiling. Importantly, they found that hVISA isolates were associated with an increased MIC of vancomycin.

The authors found that the hVISA phenotype was more common among patients with healthcare-associated infections, in those who had been previously treated with vancomycin, in those treated with medications for a MRSA infection, and in those previously treated with vancomycin specifically. Interestingly, those with an hVISA isolate in the entire cohort were less likely to die at 90 days than those with a non-hVISA isolate using a Cox regression analysis (Table 4), but among those treated with vancomycin for the index infection, those with an hVISA isolates were more than twice as likely to die by 90 days (Table 5). Those with an elevated vancomycin MIC were much more likely to have hVISA; for each increase of 1.0 in vancomycin MIC, the risk of hVISA increased by 15 times. In addition, hVISA patients had longer durations of bacteremia and survivors of the index infections were five times more likely to have a relapse of the infection by 90 days. These findings, especially the mortality outcomes, suggest that hVISA may be less virulent than non-hVISA strains in the setting of bacteremia. However, perhaps vancomycin should be avoided in patients with hVISA bacteremia. The authors went on to identify an optimal cut-off for PAP-area under the curve (AUC) that is associated with poorer clinical outcomes – 0.65. Overall the manuscript is well written and well designed. The cohort is larger than those included in previous similar studies. Unlike many previous studies, the authors importantly accounted for 1) vancomycin MIC and 2) treatment of subjects with vancomycin for the index infection applying vancomycin use as an interaction term with hVISA phenotype in their primary analyses.

I have the following questions and comments for the authors:

1. It is possible that there is an association of the hVISA phenotype and the genotype of *S. aureus* strains (as the authors imply in Lines 243-245). It may be beyond the scope of the present manuscript, but are the authors able to present spa typing, multilocus sequence typing, or whole genome sequence data for the studied MRSA isolates? It would be quite interesting and also of epidemiologic importance to identify strain types that are likely to be phenotypically hVISA.
2. If the authors could provide whole genome sequence data for the MRSA isolates studied in this manuscript, it is possible that a GWAS or a targeted search for genes previously known to be associated with vancomycin resistance (e.g., VRAR/S) may reveal genomic correlates with the hVISA phenotype. Again, this may be beyond the scope of the present study.
3. Lines 91-98. At a large academic medical center, I would expect a greater number of patients with MRSA bacteremia to present for care during a period of > 15 years. Did the studied cohort of 842 subjects include sequential patients at the center or were some relevant patients excluded?
4. Lines 99-107: Why were 2 different methods of hVISA detection used? Is it possible that in those only tested by a single method, additional testing would have yielded a different result? In what percentage of cases tested by both methods were discordant results found?
5. The authors note that other studies have found lower growth rates in clinical hVISA strains that may reflect decreased virulence in the course of clinical infections. This raises the question of whether it is a reduced growth rate generally, rather than hVISA phenotype, that is independently associated with better outcomes of bacteremia. Can the authors report growth rates of their clinical isolates? If this I beyond the scope of the present study, perhaps the authors may note this as a limitation of their study.
6. The authors do not account, among patients treated with vancomycin, for the clinical therapeutic drug monitoring data to assess whether differences associated with vancomycin therapy are, in fact, explained by appropriateness of vancomycin dosing. Is it possible that poorer outcomes among patients treated with IV vancomycin are associated with underdosing of this drug, which is notoriously difficult to dose appropriately?

Minor question.

1. Were cases included in the studied cohort all monomicrobial infections with *S. aureus*? Were those with other co-pathogens in blood excluded?

Version 1:

Reviewer comments:

Reviewer #2

(Remarks to the Author)

The authors have responded well to nearly all of my comments and questions on the previous version of the manuscript. They have specifically addressed each of the limitations clearly in the Discussion.

I have only the following questions for the authors remaining for the present manuscript:

1. Can the author please use a standardized MLST nomenclature to identify multilocus sequence types (such as those used in PubMLST [PubMLST.org])? Now in Table 4 and Line 166-177 in the Results section use the sequence numbers, but in combination, these are proxies for standardized sequence types of *S. aureus*, which are widely used in the literature, from Korea as well as in the rest of the world. If they are not able to use standardized MLST strain typing, this is a weakness that should be noted.

RESPONSE TO REVIEWER COMMENTS

REVIEWER COMMENTS

Reviewer #1 (Remarks to the Author):

The authors of “Vancomycin heteroresistant (hVISA) MRSA infections: risk factors, effects on vancomycin therapy and a clinically derived threshold for its detection” report a study exploring the clinical relevance of the hVISA phenotype. They collected clinical and laboratory data from 842 patients with MRSA bacteremia and using single and multivariate analyses explored if hVISA was correlated with worse disease outcomes – with a focus on disentangling the impact of hVISA alone from the variety of other features that contribute to disease outcome. Further, the authors explored what features were most predictive of hVISA strains, which could possibly be used as a diagnostic marker given that confirmatory lab testing of hVISA using reference methods remains a labour-intensive.

The manuscript is well written and succinct – almost too succinct - and inclusion of more detail could improve the text (see below). The laboratory methods for detection of hVISA and the statistical analysis is appropriate. Project ethics have been obtained. The data and detailed analysis parameters have been included in Supp Table 1. The manuscript was provided without line numbers.

Main comments

More detail is needed in the methods/results to understand the study hospital and therefore the context of the reported findings. In table 2, the 90-day mortality is reported as 68% and 70% for hVISA/MRSA cohorts. Is this correct? This value is high compared to a number of studies of MRSA bacteraemia (meta-analysis doi:10.1016/j.cmi.2022.03.015). Can the authors please provide more detail about the patient population included in the study and a potential explanation for these high values.

We apologize for the confusion we were in fact reporting 90 day survival in the table instead of 90 day mortality. We are very grateful of you pointing it out, it has now been changed in the Table

As for the patient population our cohort of MRSA bacteremia was derived from the largest tertiary-care hospital in South Korea, which serves as a national referral center and therefore provides a population broadly representative of cases across the country. In addition, we analyzed data from a prospective observational cohort that was established around formal Infectious Diseases consultations for MRSA bacteremia, thereby ensuring systematic case identification and consistent clinical assessment.

We have added text in our population description in the materials and methods lines 90-100.

Further, hVISA has often been linked with persistent bacteraemia. So how is “duration of bacteraemia” defined in the study hospital? Please explain.

Thank you for this valuable comment.

There have been studies to evaluate persistent bacteremia (PB) in patients with *Staphylococcus aureus* bacteremia (SAB), and we believe that there are two definitions that have been often adopted in the literature. The adjusted duration of SAB refers to the duration of SAB while on active antibiotic therapy [Kuehl R et al. *Lancet Infect Dis* 2020;20:1409-1417, Holland TL et al. *Clin Infect Dis* 2022;75:1668-1674], while the non-adjusted duration of SAB refers to the absolute duration of SAB calculated from the initial blood culture [Minejima E et al. *Clin Infect Dis* 2020;70:566-573, Tubiana S et al. *J Infect* 2016;72:544-553].

In the present study, we defined duration of bacteremia as the adjusted duration of SAB, which may better reflect the biological persistence of bacteremia despite antibiotic therapy.

We believe that adjusted duration of SAB is a continuous and time-dependent variable that provides a great statistical power and highlight a clinically meaningful dimension. Our models indicate that patients with hVISA bacteremia under vancomycin therapy experienced nearly two additional days of bacteremia compared with non-hVISA cases, which underscores the important perspective as it may exceed the predefined threshold of persistent bacteremia ($\geq 2\sim 3$ days after the initial positive blood culture) [Minejima E et al. *Clin Infect Dis* 2020;70:566-573, Tubiana S et al. *J Infect* 2016;72:544-553].

We have now amended our materials and methods (lines 123-126) to reflect that with the following text and these additional references

Kuehl R et al. Lancet Infect Dis 2020;20:1409-1417 : *Holland TL et al. Clin Infect Dis* 2022;75:1668-1674 : *Balk R et al. Chest* 1992;101:1644-1655 :

I would also recommend (in table 2) providing the full ranges for the values alongside or in place of the 25/75 percentiles.

We have now changed Table 2 to reflect range in the appropriate metrics instead of CI

The authors have not provided any information or discussion on the role of bacterial genetics in influencing clinical outcomes (other than briefly that hVISA has been linked with reduced disease severity). Although linking key bacterial genotypes with clinical outcomes is an emerging area of research – there is a number of papers that have correlated the hVISA phenotype with key mutations (reviews doi: 10.1016/j.jgar.2014.04.006 & 10.1128/CMR.00042-09), and also demonstrated that certain MRSA lineages, such as the HA-MRSA lineage ST239, has evolved towards higher VAN MICs (doi: 10.1128/mBio.00080-15). While a lot remains unknown in this area, the literature strongly suggest that some MRSA lineages have a greater predisposition to become hVISA. Therefore, understanding local molecular epidemiology of MRSA is important for understanding risk factors for hVISA. The author have captured information on MLST for all isolates (Supp Table) but do not appear to have considered this feature in their analyses. An analysis that looks into this feature, or at a minimum, explanation of why is has not been considered should be added.

We would like to thank the reviewer for this valuable suggestion. We agree that the role of MRSA lineages and their predisposition to worse clinical outcomes as well as the hVISA phenotype is indeed a valuable field in need of more research . On our part, we have now performed an additional relative risk analysis looking into specifically which sequence types from our collection should be considered a risk factor for hVISA heteroresistance. The full analysis results can be found in lines 162-169 and discussed in lines 258-264. Briefly, 45 MLSTs were present in our collection 8 of which showed to have either a significant increase

or decrease in the risk of an isolate being hVISA. We believe that this further illustrates your point that there is indeed a correlation between different Sequence types and prevalence of vancomycin heteroresistance

In terms of VAN MICs, please include more information – or even a graph/table – to describe the distribution of MIC from the hVISA and MRSA cohorts. Even an ecological cut-off value (or similar) for the hVISA phenotype in this study would be useful.

We have now created supplementary figure 1, showing the distributions of VAN MICs across the two different populations as suggested. Further statistical testing of the 2 distribution reveals that they are indeed significantly different, which further supports the observation of the multivariate models that VAN MIC is a strong risk factor for hVISA. Text describing these results has been added in lines 158-161.

Supplementary Figure 1: The distribution of VAN MIC values performed calculated by broth microdilution (mg/L) for hVISA strains (black) and VSSA strains (grey). The distributions are statistically different (t test p=0.001) indicating that the VAN MIC is an influencing factor to a strain being heteroresistant to Vncomycin.

Minor changes

1. At the start of the introduction, please add the MRSA acronym for consistency with other acronym use

Done

2. In the methods, please provide the name of the funding body that supported this work.

Added, lines 484-486.

3. Table 3 – I suggest showing the OR information in a graph format

We have now added Figure 1 showing the information in graph format, we have retained table 3 as to more thoroughly show all p values of the association to illustrate a clearer picture of the multivariate models output than can fit in a figure

Figure 2 Odds ratios of Risk factors associated with hVISA. Shown as value with 95% CI, * denotes $0.001 < p < 0.05$ and *** denotes $p < 0.001$ (multivariate logistic regression)

4. In Supp Table 1 – (in AUC tab) the results for the PAP are recorded under “Modified PAP” – if modification have been made the reference PAP procedure, please described these changes in the methods. If not, please amend the column header.
Fixed the column header

Reviewer #2 (Remarks to the Author):

Thank you for the opportunity to review the manuscript, “Vancomycin heteroresistant (hVISA) MRSA infections: risk factors, effects on vancomycin therapy and a clinically derived threshold for its detection.” Vancomycin is now the first-line therapy for MRSA bacteremia, and it is essential that we can identify any subgroup of patients with this high-mortality infection who have poorer outcome with vancomycin as alternative therapies are available. The authors of the present a report on 842 adult patients with MRSA bacteremia in South Korea, and for each they collected extensive clinical data and assessed the presence of an hVISA isolate. They compared hVISA with non-hVISA infections to assess clinical differences in the patients presenting with infection and then its impact on clinical outcomes.

Among the 842 MRSA isolates, the authors found that 22% were hVISA by population analysis profile (PAP) or by spiral gradient profiling. Importantly, they found that hVISA isolates were associated with an increased MIC of vancomycin.

The authors found that the hVISA phenotype was more common among patients with healthcare-associated infections, in those who had been previously treated with vancomycin, in those treated with medications for a MRSA infection, and in those previously treated with vancomycin specifically. Interestingly, those with an hVISA isolate in the entire cohort were less likely to die at 90 days than those with a non-hVISA isolate using a Cox regression analysis (Table 4), but among those treated with vancomycin for the index infection, those with an hVISA isolates were more than twice as likely to die by 90 days (Table 5). Those with an elevated vancomycin MIC were much more likely to have hVISA; for each increase of 1.0 in vancomycin MIC, the risk of hVISA increased by 15 times. In addition, hVISA patients had longer durations of bacteremia and survivors of the index infections were five times more likely to have a relapse of the infection by 90 days. These findings, especially the mortality outcomes, suggest that hVISA may be less virulent than non-hVISA strains in the setting of bacteremia. However, perhaps vancomycin should be avoided in patients with hVISA bacteremia. The authors went on to identify an optimal cut-off for PAP-area under the curve (AUC) that is associated with poorer clinical outcomes – 0.65.

Overall the manuscript is well written and well designed. The cohort is larger than those included in previous similar studies. Unlike many previous studies, the authors importantly accounted for 1) vancomycin MIC and 2) treatment of subjects with vancomycin for the index infection applying vancomycin use as an interaction term with hVISA phenotype in their primary analyses.

I have the following questions and comments for the authors:

1. It is possible that there is an association of the hVISA phenotype and the genotype of *S. aureus* strains (as the authors imply in Lines 243-245). It may be beyond the scope of the present manuscript, but are the authors able to present spa typing, multilocus sequence typing, or whole genome sequence data for the studied MRSA isolates? It would be quite interesting and also of epidemiologic importance to identify strain types that are likely to be phenotypically hVISA.

We agree and thank the reviewer for raising this point. Since we do not have WGS data on the collection we have now performed an additional relative risk analysis looking into specifically which sequence types from our collection should be considered a risk factor for hVISA heteroresistance. The full analysis results can be found in lines 162-169 and discussed in lines 258-264. Briefly, 45 MLSTs were present in our collection 8 of which showed to have either a significant increase or decrease in the risk of an isolate being hVISA. We believe that this further illustrates your point that there is indeed a correlation between different Sequence types and prevalence of vancomycin heteroresistance.

2. If the authors could provide whole genome sequence data for the MRSA isolates studied in this manuscript, it is possible that a GWAS or a targeted search for genes previously known to be associated with vancomycin resistance (e.g., VRAR/S) may reveal genomic correlates with the hVISA phenotype. Again, this may be beyond the scope of the present study.

We agree that a GWAS analysis of this collection could yield important insight. The collection however has not been sequenced yet. We raise the point in our discussion lines 315-321 and should funding allow this would be within the scope of future studies on this collection.

3. Lines 91-98. At a large academic medical center, I would expect a greater number of patients with MRSA bacteremia to present for care during a period of > 15 years. Did the studied cohort of 842 subjects include sequential patients at the center or were some relevant patients excluded?

Thank you for pointing out this concern.

During the study period, we initially identified 1,592 adult patients (≥ 18 years old) with a first episode of MRSA bacteremia. There were 264 patients who met the exclusion criteria that includes 207 patients with polymicrobial bacteremia and 57 patients who discharged before positive blood culture results. After applying for the exclusion criteria, 1,328 patients remained for the initial analysis. Among them, 842 stored MRSA bacteremia isolates underwent PAP testing to determine hVISA status.

To further clarify this we have made clarifying changes to the manuscript. in the materials and methods lines 90-100 and in the results lines 139-144

4. Lines 99-107: Why were 2 different methods of hVISA detection used?

We understand the confusion. At the beginning in 2008, the experiments to detect hVISA were performed manually, because we did not have access to the spiral plate instrument. After acquiring the equipment in 2011, we conducted the experiments using the spiral plater. The experiments and subsequent analyses were consistently performed by the same researcher throughout the study period since 2008. The PAP assays with the spiral plater were retrospectively performed for earlier isolates collected between 2008 and 2011. When discrepancies arose between results obtained by the two different methods, the results generated by the spiral plater were preferentially considered.

Is it possible that in those only tested by a single method, additional testing would have yielded a different result? In what percentage of cases tested by both methods were discordant results found?

Thank you for this very interesting suggestion. We have now performed a comparative analysis between the 2 methods and included it in the materials and methods section lines 112-114.

“Out of 65 VSSA isolates tested with both methods 30% (20) show disagreement between methods and 70% (45) agreement. Out of 31 hVISA tested with both methods 38% (12) show disagreement between the methods 61% (19) agreement.”

The numbers of strains tested with both methods is unfortunately not high enough to be able to conclusively investigate if testing with the other method would have yielded a different result. However this analysis shows the need for consistency in clinical testing of hVISA and perhaps new diagnostics, a point we have now included in our discussion lines 265-268.

5. The authors note that other studies have found lower growth rates in clinical hVISA strains that may reflect decreased virulence in the course of clinical infections. This raises the question of whether it is a reduced growth rate generally, rather than hVISA phenotype, that is independently associated with better outcomes of bacteremia. Can the authors report growth rates of their clinical isolates? If this is beyond the scope of the present study, perhaps the authors may note this as a limitation of their study.

Thank you for this comment. We agree it is very complicated to disentangle the underlying cause of hVISA strains having lower mortality when not treated with VAN compared to other

MRSAs. The reduced growth rate alone could not explain this in our opinion so at this point one can only speculate on potential causes. We have amended our discussion (lines 315-321) to reflect that reduced virulence and growth are viable explanations of this and to state that future studies, dedicated to exploring exactly the mechanisms behind this, should be performed with complete in vitro competition and growth rate experiments.

6. The authors do not account, among patients treated with vancomycin, for the clinical therapeutic drug monitoring data to assess whether differences associated with vancomycin therapy are, in fact, explained by appropriateness of vancomycin dosing. Is it possible that poorer outcomes among patients treated with IV vancomycin are associated with underdosing of this drug, which is notoriously difficult to dose appropriately?

Thank you for raising this point. Unfortunately in our collection, since it spans almost 2 decades, not all patients received TDM. Therefore we deemed TDM a confounder in the effects of hVISA specifically because we could not account for vancomycin dosing in a large part of our cohort. TDM is well documented to improve patient outcomes and help mitigate Vancomycin`s dosing problems so we have now included text in our discussion (lines 306-307) indicating that appropriate TDM is one of the possible practices that can help mitigate the effects of the hVISA phenotype as well

Minor question.

1. Were cases included in the studied cohort all monomicrobial infections with *S. aureus*? Were those with other co-pathogens in blood excluded?

Patients with monomicrobial MRSA bacteremia were included in the present study, and cases with polymicrobial bacteremia were excluded. The above information can be also be found in Materials and methods lines 90-100.

Response to reviewer 2

Reviewer #2 (Remarks to the Author)

The authors have responded well to nearly all of my comments and questions on the previous version of the manuscript. They have specifically addressed each of the limitations clearly in the Discussion.

I have only the following questions for the authors remaining for the present manuscript:

1. Can the author please use a standardized MLST nomenclature to identify multilocus sequence types (such as those used in PubMLST [PubMLST.org])? Now in Table 4 and Line 166-177 in the Results section use the sequence numbers, but in combination, these are proxies for standardized sequence types of *S. aureus*, which are widely used in the literature, from Korea as well as in the rest of the world. If they are not able to use standardized MLST strain typing, this is a weakness that should be noted.

Thanks for pointing this out. We have now added the standard sequence types numbers in Table 4 and lines 115, 117 (in converted pdf).